# G-EVAL: NLG Evaluation using GPT-4 with Better Human Alignment

**Yang Liu**   **Dan Iter**   **Yichong Xu**
**Shuohang Wang**   **Ruochen Xu**   **Chenguang Zhu**

Microsoft Azure AI
*yaliu10@microsoft.com*

## Abstract

The quality of texts generated by natural language generation (NLG) systems is hard to measure automatically. Conventional reference-based metrics, such as BLEU and ROUGE, have been shown to have relatively low correlation with human judgments, especially for tasks that require creativity and diversity. Recent studies suggest using large language models (LLMs) as reference-free metrics for NLG evaluation, which have the benefit of being applicable to new tasks that lack human references. However, these LLM-based evaluators still have lower human correspondence than medium-size neural evaluators. In this work, we present G-EVAL, a framework of using large language models with chain-of-thoughts (CoT) and a form-filling paradigm, to assess the quality of NLG outputs. We experiment with two generation tasks, text summarization and dialogue generation. We show that G-EVAL with GPT-4 as the backbone model achieves a Spearman correlation of $0.514$ with human on summarization task, outperforming all previous methods by a large margin. We also propose analysis on the behavior of LLM-based evaluators, and highlight the potential concern of LLM-based evaluators having a bias towards the LLM-generated texts. [1]

## 1   Introduction

Evaluating the quality of natural language generation systems is a challenging problem even when large language models can generate high-quality and diverse texts that are often indistinguishable from human-written texts (Ouyang et al., 2022). Traditional automatic metrics, such as BLEU (Papineni et al., 2002), ROUGE (Lin, 2004), and METEOR (Banerjee and Lavie, 2005), are widely used for NLG evaluation, but they have been shown to have relatively low correlation with human judgments, especially for open-ended generation tasks.

Moreover, these metrics require associated reference output, which is costly to collect for new tasks.

Recent studies propose directly using LLMs as reference-free NLG evaluators (Fu et al., 2023; Wang et al., 2023a). The idea is to use the LLMs to score the candidate output based on its generation probability without any reference target, under the assumption that the LLMs have learned to assign higher probabilities to high-quality and fluent texts. Meanwhile, it is becoming popular to use more powerful LLMs like GPT-4 to evaluate smaller or student models, like in Alpaca (Taori et al., 2023) and Vicuna (Zheng et al., 2023). However, the validity and reliability of using LLMs as NLG evaluators have not been systematically investigated. In addition, meta-evaluations show that these LLM-based evaluators still have lower human correspondence than medium-size neural evaluators (Zhong et al., 2022). Thus, there is a need for a more effective and reliable framework for using LLMs for NLG evaluation.

In this paper, we propose G-EVAL, a framework of using LLMs with chain-of-thoughts (CoT) (Wei et al., 2022) to evaluate the quality of generated texts in a form-filling paradigm. By only feeding the Task Introduction and the Evaluation Criteria as a prompt, we ask LLMs to generate a CoT of detailed Evaluation Steps. Then we use the prompt along with the generated CoT to evaluate the NLG outputs. The evaluator output is formatted as a form. Moreover, the probabilities of the output rating tokens can be used to refine the final metric. We conduct extensive experiments on three meta-evaluation benchmarks of two NLG tasks: text summarization and dialogue generation. The results show that G-EVAL can outperform existing NLG evaluators by a large margin in terms of correlation with human evaluations. Finally, we conduct analysis on the behavior of LLM-based evaluators, and highlight the potential issue of LLM-based evaluator having a bias towards the LLM-generated

---

[1] https://github.com/nlpyang/geval

texts.

To summarize, our main contributions and findings in this paper are:

1. G-EVAL generally outperforms reference-based and reference-free baseline metrics in terms of correlation with human quality judgments, especially for open-ended and creative NLG tasks, such as dialogue response generation.

2. We propose to use automatic chain-of-thought to improve the performance of LLM-based evaluators by providing more context and guidance.

3. We propose to re-weight the discrete scores by their respective token probabilities to provide a more fine-grained continuous score for G-EVAL.

4. We conduct an analysis of the potential issue that LLM-based metrics have a preference of LLM-generated texts over human-written texts, which may lead to the self-reinforcement of LLMs if LLM-based metrics are used as the reward signal for improving themselves.

## 2 Method

G-EVAL is a prompt-based evaluator with three main components: 1) a prompt that contains the definition of the evaluation task and the desired evaluation criteria, 2) a chain-of-thoughts (CoT) that is a set of intermediate instructions generated by the LLM describing the detailed evaluation steps, and 3) a scoring function that calls LLM and calculates the score based on the probabilities of the return tokens.

**Prompt for NLG Evaluation** The prompt is a natural language instruction that defines the evaluation task and the desired evaluation criteria. For example, for text summarization, the prompt can be:

> *You will be given one summary written for a news article. Your task is to rate the summary on one metric.*
>
> *Please make sure you read and understand these instructions carefully. Please keep this document open while reviewing, and refer to it as needed.*

The prompt should also contain customized evaluation criteria for different NLG tasks and, such as coherence, conciseness, or grammar. For example, for evaluating coherence in text summarization, we add the following content to the prompt:

> *Evaluation Criteria:*
>
> *Coherence (1-5) - the collective quality of all sentences. We align this dimension with the DUC quality question of structure and coherence whereby "the summary should be well-structured and well-organized. The summary should not just be a heap of related information, but should build from sentence to sentence to a coherent body of information about a topic."*

**Auto Chain-of-Thoughts for NLG Evaluation** The chain-of-thoughts (CoT) is a sequence of intermediate representations that are generated by the LLM during the text generation process. For evaluation tasks, some criteria need a more detailed evaluation instruction beyond the simple definition, and it is time-consuming to manually design such evaluation steps for each task. We find that LLM can generate such evaluation steps by itself. The CoT can provide more context and guidance for the LLM to evaluate the generated text, and can also help to explain the evaluation process and results. For example, for evaluating coherence in text summarization, we add a line of *"Evaluation Steps:"* to the prompt and let LLM to generate the following CoT automatically:

> *1. Read the news article carefully and identify the main topic and key points.*
>
> *2. Read the summary and compare it to the news article. Check if the summary covers the main topic and key points of the news article, and if it presents them in a clear and logical order.*
>
> *3. Assign a score for coherence on a scale of 1 to 5, where 1 is the lowest and 5 is the highest based on the Evaluation Criteria.*

**Scoring Function** The scoring function calls the LLM with the designed prompt, auto CoT, the input context and the target text that needs to be evaluated. Unlike GPTScore (Fu et al., 2023) which uses

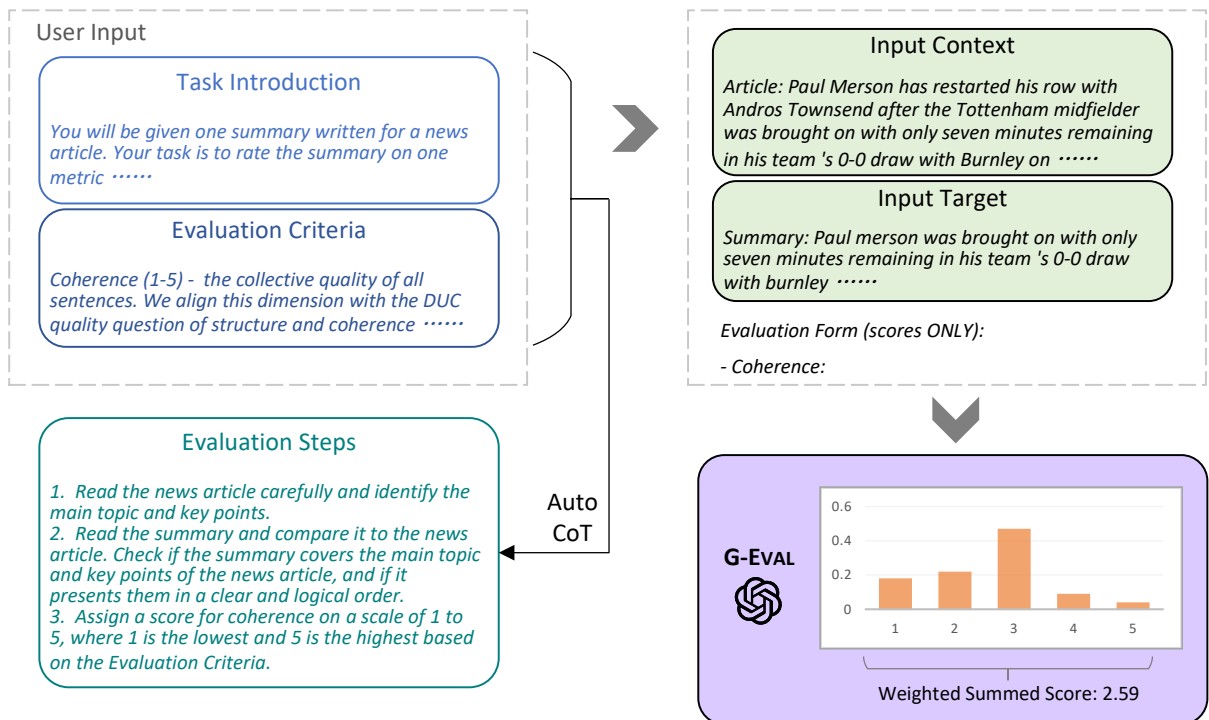

Figure 1: The overall framework of G-EVAL. We first input Task Introduction and Evaluation Criteria to the LLM, and ask it to generate a CoT of detailed Evaluation Steps. Then we use the prompt along with the generated CoT to evaluate the NLG outputs in a form-filling paradigm. Finally, we use the probability-weighted summation of the output scores as the final score.

the conditional probability of generating the target text as an evaluation metric, G-EVAL directly performs the evaluation task with a form-filling paradigm. This provides a more flexible way to evaluate the text as the model can behave directly based on the evaluation criteria and steps. For example, for evaluating coherence in text summarization, we concatenate the prompt, the CoT, the news article, and the summary, and then call the LLM to output a score from 1 to 5 for each evaluation aspect, based on the defined criteria.

However, we notice this direct scoring function has two issues:

1. For some evaluation tasks, one digit usually dominates the distribution of the scores, such as 3 for a 1 - 5 scale. This may lead to the low variance of the scores and the low correlation with human judgments.

2. LLMs usually only output integer scores, even when the prompt explicitly requests decimal values. This leads to many ties in evaluation scores which do not capture the subtle difference between generated texts.

To address these issues, we propose using the

probabilities of output tokens from LLMs to normalize the scores and take their weighted summation as the final results. Formally, given a set of scores (like from 1 to 5) predefined in the prompt $S = \{s_1, s_2, ..., s_n\}$, the probability of each score $p(s_i)$ is calculated by the LLM, and the final score is:

$$score = \sum_{i=1}^{n} p(s_i) \times s_i \qquad (1)$$

This method obtains more fine-grained, continuous scores that better reflect the quality and diversity of the generated texts.

## 3 Experiments

Following Zhong et al. (2022), we meta-evaluate our evaluator on three benchmarks, SummEval, Topical-Chat and QAGS, of two NLG tasks, summarization and dialogue response generation.

### 3.1 Implementation Details

We use OpenAI's GPT family as our LLMs, including GPT-3.5 (text-davinci-003) and GPT-4. For GPT-3.5, we set decoding temperature to 0 to increase the model's determinism. For GPT-4, as it

| Metrics | Coherence | | Consistency | | Fluency | | Relevance | | AVG | |
|---|---|---|---|---|---|---|---|---|---|---|
| | $\rho$ | $\tau$ | $\rho$ | $\tau$ | $\rho$ | $\tau$ | $\rho$ | $\tau$ | $\rho$ | $\tau$ |
| ROUGE-1 | 0.167 | 0.126 | 0.160 | 0.130 | 0.115 | 0.094 | 0.326 | 0.252 | 0.192 | 0.150 |
| ROUGE-2 | 0.184 | 0.139 | 0.187 | 0.155 | 0.159 | 0.128 | 0.290 | 0.219 | 0.205 | 0.161 |
| ROUGE-L | 0.128 | 0.099 | 0.115 | 0.092 | 0.105 | 0.084 | 0.311 | 0.237 | 0.165 | 0.128 |
| BERTScore | 0.284 | 0.211 | 0.110 | 0.090 | 0.193 | 0.158 | 0.312 | 0.243 | 0.225 | 0.175 |
| MOVERSscore | 0.159 | 0.118 | 0.157 | 0.127 | 0.129 | 0.105 | 0.318 | 0.244 | 0.191 | 0.148 |
| BARTScore | 0.448 | 0.342 | 0.382 | 0.315 | 0.356 | 0.292 | 0.356 | 0.273 | 0.385 | 0.305 |
| UniEval | 0.575 | 0.442 | 0.446 | 0.371 | 0.449 | 0.371 | 0.426 | 0.325 | 0.474 | 0.377 |
| GPTScore | 0.434 | – | 0.449 | – | 0.403 | – | 0.381 | – | 0.417 | – |
| G-EVAL-3.5 | 0.440 | 0.335 | 0.386 | 0.318 | 0.424 | 0.347 | 0.385 | 0.293 | 0.401 | 0.320 |
| - Probs | 0.359 | *0.313* | 0.361 | *0.344* | 0.339 | *0.323* | 0.327 | *0.288* | 0.346 | *0.317* |
| G-EVAL-4 | **0.582** | **0.457** | **0.507** | **0.425** | **0.506** | **0.455** | **0.547** | **0.433** | **0.514** | **0.418** |
| - Probs | 0.560 | *0.472* | 0.501 | *0.459* | 0.505 | *0.473* | 0.511 | *0.444* | 0.502 | *0.446* |
| - CoT | 0.564 | 0.454 | 0.493 | 0.413 | 0.483 | 0.431 | 0.538 | 0.427 | 0.500 | 0.407 |
| - Description | 0.513 | 0.424 | 0.421 | 0.344 | 0.447 | 0.373 | 0.479 | 0.388 | 0.479 | 0.377 |

Table 1: Summary-level Spearman ($\rho$) and Kendall-Tau ($\tau$) correlations of different metrics on SummEval benchmark. G-EVAL without probabilities (*italicized*) should not be considered as a fair comparison to other metrics on $\tau$, as it leads to many ties in the scores. This results in a higher Kendall-Tau correlation, but it does not fairly reflect the true evaluation ability. More details are in Section 4.

does not support the output of token probabilities, we set '$n = 20, temperature = 1, top\_p = 1$' to sample 20 times to estimate the token probabilities. We use G-EVAL-4 to indicate G-EVAL with GPT-4 as the backbone model, and G-EVAL-3.5 to indicate G-EVAL with GPT-3.5 as the backbone model. Example prompts for each task are provided in the Appendix.

### 3.2 Benchmarks

We adopt three meta-evaluation benchmarks to measure the correlation between G-EVAL and human judgments.

**SummEval** (Fabbri et al., 2021) is a benchmark that compares different evaluation methods for summarization. It gives human ratings for four aspects of each summary: fluency, coherence, consistency and relevance. It is built on the CNN/DailyMail dataset (Hermann et al., 2015)

**Topical-Chat** (Mehri and Eskenazi, 2020) is a testbed for meta-evaluating different evaluators on dialogue response generation systems that use knowledge. We follow (Zhong et al., 2022) to use its human ratings on four aspects: naturalness, coherence, engagingness and groundedness.

**QAGS** (Wang et al., 2020) is a benchmark for evaluating hallucinations in the summarization task. It aims to measure the consistency dimension of summaries by asking and answering questions. It is collected from two different news summarization datasets CNN/DailyMail and XSum.

### 3.3 Baselines

We evaluate G-EVAL against various evaluators that achieved state-of-the-art performance.

**BERTScore** (Zhang et al., 2019) measures the similarity between two texts based on the contextualized embedding from BERT (Devlin et al., 2019).

**MoverScore** (Zhao et al., 2019) improves BERTScore by adding soft alignments and new aggregation methods to obtain a more robust similarity measure.

**BARTScore** (Yuan et al., 2021) is a unified evaluator which evaluate with the average likelihood of the pretrained encoder-decoder model, BART (Lewis et al., 2020). It can predict different scores depending on the formats of source and target.

**FactCC** and **QAGS** (Kryściński et al., 2020; Wang et al., 2020) are two evaluators that measure the factual consistency of generated summaries. FactCC is a BERT-based classifier that predicts whether a summary is consistent with the source document. QAGS is a question-answering based evaluator that generates questions from the summary and checks if the answers can be found in the source document.

| Metrics | Naturalness | | Coherence | | Engagingness | | Groundedness | | AVG | |
|---|---|---|---|---|---|---|---|---|---|---|
| | $r$ | $\rho$ | $r$ | $\rho$ | $r$ | $\rho$ | $r$ | $\rho$ | $r$ | $\rho$ |
| ROUGE-L | 0.176 | 0.146 | 0.193 | 0.203 | 0.295 | 0.300 | 0.310 | 0.327 | 0.243 | 0.244 |
| BLEU-4 | 0.180 | 0.175 | 0.131 | 0.235 | 0.232 | 0.316 | 0.213 | 0.310 | 0.189 | 0.259 |
| METEOR | 0.212 | 0.191 | 0.250 | 0.302 | 0.367 | 0.439 | 0.333 | 0.391 | 0.290 | 0.331 |
| BERTScore | 0.226 | 0.209 | 0.214 | 0.233 | 0.317 | 0.335 | 0.291 | 0.317 | 0.262 | 0.273 |
| USR | 0.337 | 0.325 | 0.416 | 0.377 | 0.456 | 0.465 | 0.222 | 0.447 | 0.358 | 0.403 |
| UniEval | 0.455 | 0.330 | 0.602 | 0.455 | 0.573 | 0.430 | 0.577 | 0.453 | 0.552 | 0.417 |
| G-Eval-3.5 | 0.532 | 0.539 | 0.519 | 0.544 | **0.660** | **0.691** | **0.586** | 0.567 | 0.574 | 0.585 |
| G-Eval-4 | **0.549** | **0.565** | **0.594** | **0.605** | 0.627 | 0.631 | 0.531 | 0.551 | **0.575** | **0.588** |

Table 2: Turn-level Spearman ($\rho$) and Kendall-Tau ($\tau$) correlations of different metrics on Topical-Chat benchmark.

**USR** (Mehri and Eskenazi, 2020) is evaluator that assesses dialogue response generation from different perspectives. It has several versions that assign different scores to each target response.

**UniEval** (Zhong et al., 2022) is a unified evaluator that can evaluate different aspects of text generation as QA tasks. It uses a pretrained T5 model (Raffel et al., 2020) to encode the evaluation task, source and target texts as questions and answers, and then computes the QA score as the evaluation score. It can also handle different evaluation tasks by changing the question format.

**GPTScore** (Fu et al., 2023) is a new framework that evaluates texts with generative pre-training models like GPT-3. It assumes that a generative pre-training model will assign a higher probability of high-quality generated text following a given instruction and context. Unlike G-Eval, GPTScore formulates the evaluation task as a conditional generation problem instead of a form-filling problem. We report the score of GPTScore with GPT3-text-davinci-003 as the LLM, which is also usually referred as GPT-3.5.

### 3.4 Results for Summarization

We adopt the same approach as Zhong et al. (2022) to evaluate different summarization metrics using summary-level Spearman and Kendall-Tau correlation. The first part of Table 1 shows the results of metrics that compare the semantic similarity between the model output and the reference text. These metrics perform poorly on most dimensions. The second part shows the results of metrics that use neural networks to learn from human ratings of summary quality. These metrics have much higher correlations than the similarity-based metrics, suggesting that they are more reliable for summarization evaluation.

In the last part of Table 1 which corresponds to GPT-based evaluators, GPTScore also uses GPTs for evaluating summarization texts, but relies on GPT's conditional probabilities of the given target. G-Eval substantially surpasses all previous state-of-the-art evaluators on the SummEval benchmark. G-Eval-4 achieved much higher human correspondence compared with G-Eval-3.5 on both Spearman and Kendall-Tau correlation, which indicates that the larger model size of GPT-4 is beneficial for summarization evaluation. G-Eval also outperforms GPTScore on several dimension, demonstrating the effectiveness of the simple form-filling paradigm.

### 3.5 Results for Dialogue Generation

We use the Topical-chat benchmark from Mehri and Eskenazi (2020) to measure how well different evaluators agree with human ratings on the quality of dialogue responses. We calculate the Pearson and Spearman correlation for each turn of the dialogue. Table 2 shows that similarity-based metrics have good agreement with humans on how engaging and grounded the responses are, but not on the other aspects. With respect to the learning-based evaluators, before G-Eval, UniEval predicts scores that are most consistent with human judgments across all aspects.

As shown in the last part, G-Eval also substantially surpasses all previous state-of-the-art evaluator on the Topical-Chat benchmark. Notably, the G-Eval-3.5 can achieve similar results with G-Eval-4. This indicates that this benchmark is relatively easy for the G-Eval model.

### 3.6 Results on Hallucinations

Advanced NLG models often produce text that does not match the context input (Cao et al., 2018), and

| Metrics | QAGS-CNN | | | QAGS-XSUM | | | Average | | |
|---|---|---|---|---|---|---|---|---|---|
| | $r$ | $\rho$ | $\tau$ | $r$ | $\rho$ | $\tau$ | $r$ | $\rho$ | $\tau$ |
| ROUGE-2 | 0.459 | 0.418 | 0.333 | 0.097 | 0.083 | 0.068 | 0.278 | 0.250 | 0.200 |
| ROUGE-L | 0.357 | 0.324 | 0.254 | 0.024 | -0.011 | -0.009 | 0.190 | 0.156 | 0.122 |
| BERTScore | 0.576 | 0.505 | 0.399 | 0.024 | 0.008 | 0.006 | 0.300 | 0.256 | 0.202 |
| MoverScore | 0.414 | 0.347 | 0.271 | 0.054 | 0.044 | 0.036 | 0.234 | 0.195 | 0.153 |
| FactCC | 0.416 | 0.484 | 0.376 | 0.297 | 0.259 | 0.212 | 0.356 | 0.371 | 0.294 |
| QAGS | 0.545 | - | - | 0.175 | - | - | 0.375 | - | - |
| BARTScore | **0.735** | 0.680 | 0.557 | 0.184 | 0.159 | 0.130 | 0.459 | 0.420 | 0.343 |
| CTC | 0.619 | 0.564 | 0.450 | 0.309 | 0.295 | 0.242 | 0.464 | 0.430 | 0.346 |
| UniEval | 0.682 | 0.662 | 0.532 | 0.461 | 0.488 | 0.399 | 0.571 | 0.575 | 0.465 |
| G-Eval-3.5 | 0.477 | 0.516 | 0.410 | 0.211 | 0.406 | 0.343 | 0.344 | 0.461 | 0.377 |
| G-Eval-4 | 0.631 | **0.685** | **0.591** | **0.558** | **0.537** | **0.472** | **0.599** | **0.611** | **0.525** |

Table 3: Pearson ($r$), Spearman ($\rho$) and Kendall-Tau ($\tau$) correlations of different metrics on QAGS benchmark.

recent studies find even powerful LLMs also suffer from the problem of hallucination. This motivates recent research to design evaluators for measuring the `consistency` aspect in summarization (Kryściński et al., 2020; Wang et al., 2020; Cao et al., 2020; Durmus et al., 2020). We test the QAGS meta-evaluation benchmark, which includes two different summarization datasets: CNN/DailyMail and XSum (Narayan et al., 2018) Table 3 shows that BARTScore performs well on the more extractive subset (QAGS-CNN), but has low correlation on the more abstractive subset (QAGS-Xsum). UniEval has good correlation on both subsets of the data.

On average, G-Eval-4 outperforms all state-of-the-art evaluators on QAGS, with a large margin on QAGS-Xsum. G-Eval-3.5, on the other hand, failed to perform well on this benchmark, which indicates that the consistency aspect is sensitive to the LLM's capacity. This result is consistent with Table 1.

## 4 Analysis

**Will G-Eval prefer LLM-based outputs?** One concern about using LLM as an evaluator is that it may prefer the outputs generated by the LLM itself, rather than the high-quality human-written texts. To investigate this issue, we conduct an experiment on the summarization task, where we compare the evaluation scores of the LLM-generated and the human-written summaries. We use the dataset collected in Zhang et al. (2023), where they first ask freelance writers to write high-quality summaries for news articles, and then ask annotators to compare human-written summaries and LLM-

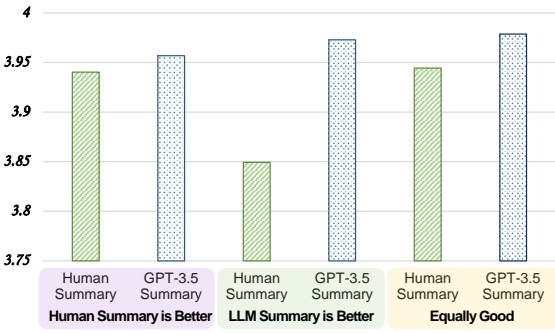

Figure 2: Averaged G-Eval-4's scores for human-written summaries and GPT-3.5 summaries, divided by human judges' preference.

generated summaries (using GPT-3.5, text-davinci-003).

The dataset can be divided in three categories: 1) human-written summaries that are rated *higher* than GPT-3.5 summaries by human judges, 2) human-written summaries that are rated *lower* than GPT-3.5 summaries by human judges, and 3) human-written summaries and GPT-3.5 summaries are rated *equally* good by human judges. We use G-Eval-4 to evaluate the summaries in each category, and compare the averaged scores. [2]

The results are shown in Figure 2. We can see that, G-Eval-4 assigns higher scores to human-written summaries when human judges also prefer human-written summaries, and assigns lower scores when human judges prefer GPT-3.5 summaries. However, G-Eval-4 always gives higher scores to GPT-3.5 summaries than human-written

---
[2] We use G-Eval-4 in this experiment, because its superiority in evaluating summarization tasks. Although it has different distribution with with GPT-3.5, the two LLMs should share similar behaviors in terms of text generation.

summaries, even when human judges prefer human-written summaries. We propose two potential reasons for this phenomenon:

1. NLG outputs from high-quality systems are in natural difficult to evaluate. The authors of the original paper found that inter-annotator agreement on judging human-written and LLM-generated summaries is very low, with Krippendorff's alpha at 0.07.

2. G-EVAL may have a bias towards the LLM-generated summaries because the model could share the same concept of evaluation criteria during generation and evaluation.

Our work should be considered as a preliminary study on this issue, and more research is needed to fully understand the behavior of LLM-based evaluators to reduce its inherent bias towards LLM-generated text. We highlight this concern in the context that LLM-based evaluators may lead to self-reinforcement of LLMs if the evaluation score is used as a reward signal for further tuning. And this could result in the over-fitting of the LLMs to their own evaluation criteria, rather than the true evaluation criteria of the NLG tasks.

**The Effect of Chain-of-Thoughts**  We compare the performance of G-EVAL with and without chain-of-thoughts (CoT) on the SummEval benchmark. Table 1 shows that G-EVAL-4 with CoT has higher correlation than G-EVAL-4 without CoT on all dimensions, especially for fluency. This suggests that CoT can provide more context and guidance for the LLM to evaluate the generated text, and can also help to explain the evaluation process and results. And it is shown that CoT is more useful on consistency and fluency dimensions. We also provide results of G-EVAL with a simple prompting baseline on SummEval (only asking GPT-4 to score a summary from 1-5 on each dimension, without detailed task introduction, evaluation criteria and CoT).

**The Effect of Probability Normalization**  We compare the performance of G-EVAL with and without probability normalization on the SummEval benchmark. Table 1 shows that, on Kendall-Tau correlation, G-EVAL-4 with probabilities is inferior to G-EVAL-4 without probabilities on SummEval. We believe this is related to the calculation of Kendall-Tau correlation, which is based on the number of concordant and discordant pairs. Direct

scoring without probabilities can lead to many ties, which are not counted as either concordant or discordant. This may result in a higher Kendall-Tau correlation, but it does not reflect the model's true capacity of evaluating the generated texts. On the other hand, probability normalization can obtain more fine-grained, continuous scores that better capture the subtle difference between generated texts. This is reflected by the higher Spearman correlation of G-EVAL-4 with probabilities, which is based on the rank order of the scores.

**The Effect of Different LLMs**  We compare the performance of G-EVAL with different LLMs on the SummEval and QAGS benchmarks. Table 1 and Table 3 show that G-EVAL-4 has higher correlation than G-EVAL-3.5 on most dimensions and datasets, except for engagingness and groundedness on the Topical-Chat benchmark. This demonstrates that a better LLM can improve the performance of G-EVAL, especially for more challenging and complex evaluation tasks, such as consistency and relevance.

## 5  Related Work

**Ngram-based Metrics**  Ngram-based metrics refer to the scores for evaluating the NLG models by measuring the lexical overlap between a generated text and a reference text. BLEU (Papineni et al., 2002) is the most widely used metric for machine translation evaluation, which calculates the geometric mean of modified n-gram precision and a brevity penalty. ROUGE (Lin, 2004) is a recall-oriented metric for summarization evaluation, which measures the n-gram overlap between a generated summary and a set of reference summaries. It has been shown that more than 60% of recent papers on NLG only rely on ROUGE or BLEU to evaluate their systems (Kasai et al., 2022). However, these metrics fail to measure content quality (Reiter and Belz, 2009) or capture syntactic errors (Stent et al., 2005), and therefore do not reflect the reliability of NLG systems accurately.

**Embedding-based Metrics**  Embedding-based metrics refer to the scores for evaluating the NLG models by measuring the semantic similarity between a generated text and a reference text based on the word or sentence embeddings. WMD (Kusner et al., 2015) is a metric that measures the distance between two texts based on the word embeddings. BERTScore (Zhang et al., 2019) measures

the similarity between two texts based on the contextualized embedding from BERT (Devlin et al., 2019). MoverScore (Zhao et al., 2019) improves BERTScore by adding soft alignments and new aggregation methods to obtain a more robust similarity measure. (Clark et al., 2019) propose a metric that evaluates multi-sentence texts by computing the similarity between the generated text and the reference text based on the sentence embeddings.

**Task-specific Evaluators**    Task-specific metrics refer to the scores for evaluating the NLG models by measuring the quality of the generated texts based on the specific task requirements. For example, summarization tasks need to assess the `consistency` of the generated summaries (Kryściński et al., 2020; Wang et al., 2020; Cao et al., 2020; Durmus et al., 2020), and dialogue response generation tasks need to assess the `coherence` of the generated responses (Dziri et al., 2019; Ye et al., 2021; Ghazarian et al., 2019). However, these metrics are not generalizable to other NLG tasks, and they are not able to measure the overall quality of the generated texts.

**Unified Evaluators**    Recently, some evaluators have been developed to assess text quality from multiple dimensions by varying the input and output contents (Yuan et al., 2021) or the model variants (Mehri and Eskenazi, 2020) they use. UniEval (Zhong et al., 2022) is a unified evaluator that can evaluate different aspects of text generation as QA tasks. By changing the question format, it can handle different evaluation tasks.

**LLM-based Evaluators**    Fu et al. (2023) propose GPTScore, a new framework that evaluated texts with generative pre-training models like GPT-3. It assumes that a generative pre-training model will assign a higher probability of high-quality generated text following a given instruction and context. Wang et al. (2023a) conduct a preliminary survey of using ChatGPT as a NLG evaluator. Kocmi and Federmann (2023); Lu et al. (2023) proposed to use GPT models for evaluating machine translation tasks. Very recently, Wang et al. (2023b) investigated the problem of unfairness when using large models in evaluating dialogue responses.

## 6    Conclusion

In this paper, we propose G-EVAL, a framework of using LLM with chain-of-thoughts (CoT) to evaluate the quality of generated texts. We conduct extensive experiments on two NLG tasks, text summarization and dialogue generation, and show that G-EVAL can outperform state-of-the-art evaluators and achieve higher human correspondence. We also propose preliminary analysis on the behavior of LLM-based evaluators, and highlight the potential issue of LLM-based evaluator having a bias towards the LLM-generated texts. We hope our work can inspire more research on using LLMs for NLG evaluation, and also raise awareness of the potential risks and challenges of using LLMs as evaluators.

## Limitations

G-EVAL is a framework that uses LLMs to evaluate the quality of generated texts. However, it also has some limitations that need to be addressed in future work.

1. As we already discussed in the paper, G-EVAL may have a bias towards the LLM-generated texts. This may lead to the self-reinforcement of LLMs if the evaluation score is used as a reward signal for further tuning. And this could result in the over-fitting of the LLMs to their own evaluation criteria, rather than the true evaluation criteria of the NLG tasks.

2. G-EVAL is limited by the availability and accessibility of LLMs. Currently, most LLMs are not publicly available, and require special access or payment to use. This may limit the applicability and reproducibility of G-EVAL. Moreover, the LLMs are constantly updated, which may lead to inconsistent evaluation results across different versions of the LLMs.

3. We meta-evaluate G-EVAL on two NLG tasks, text summarization and dialogue generation. However, there are some emerging NLG tasks in the LLM era where users prompt with free-form natural language instructions. In this case, the evaluation criteria may need to be more flexible and adaptive to the user's intention and preference. Therefore, more research is needed to explore how to use G-EVAL for evaluating these new types of NLG tasks.

## Ethics Statement

The G-EVAL framework we proposed is designed to offer a more effective and reliable method for assessing natural language generation systems. Its

purpose is to aid researchers, developers, and other interested parties in evaluating the quality of text produced by NLG systems. Possible risks could exist if G-EVAL is unable to precisely evaluate the quality of produced texts or shows a preference for LLM-created texts. This could lead to developers overestimating the performance of their systems or unintentionally reinforcing biases in their models. Furthermore, users depending on the generated material may receive low-quality or biased information.

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

## A  Example Prompts

**Evaluate Coherence in the Summarization Task**

*You will be given one summary written for a news article.*

*Your task is to rate the summary on one metric.*

*Please make sure you read and understand these instructions carefully. Please keep this document open while reviewing, and refer to it as needed.*

*Evaluation Criteria:*

*Coherence (1-5) - the collective quality of all sentences. We align this dimension with the DUC quality question of structure and coherence whereby "the summary should be well-structured and well-organized. The summary should not just be a heap of related information, but should build from sentence to sentence to a coherent body of information about a topic."*

*Evaluation Steps:*

*1. Read the news article carefully and identify the main topic and key points.*

*2. Read the summary and compare it to the news article. Check if the summary covers the main topic and key points of the news article, and if it presents them in a clear and logical order.*

*3. Assign a score for coherence on a scale of 1 to 5, where 1 is the lowest and 5 is the highest based on the Evaluation Criteria.*

*Example:*

*Source Text:*

*{{Document}}*

*Summary:*

*{{Summary}}*

*Evaluation Form (scores ONLY):*

*- Coherence:*

## Evaluate Engagingness in the Dialogue Generation Task

*You will be given a conversation between two individuals. You will then be given one potential response for the next turn in the conversation. The response concerns an interesting fact, which will be provided as well.*

*Your task is to rate the responses on one metric.*

*Please make sure you read and understand these instructions carefully. Please keep this document open while reviewing, and refer to it as needed.*

*Evaluation Crieteria:*

*Engagingness (1-3) Is the response dull/interesting?*

*- A score of 1 (dull) means that the response is generic and dull.*

*- A score of 2 (somewhat interesting) means the response is somewhat interesting and could engage you in the conversation (e.g., an opinion, thought)*

*- A score of 3 (interesting) means the response is very interesting or presents an interesting fact*

*Evaluation Steps:*

*1. Read the conversation, the corresponding fact and the response carefully.*

*2. Rate the response on a scale of 1-3 for engagingness, according to the criteria above.*

*3. Provide a brief explanation for your rating, referring to specific aspects of the response and the conversation.*

*Example:*

*Conversation History:*

*{{Document}}*

*Corresponding Fact:*

*{{Fact}}*

*Response:*

*{{Response}}*

*Evaluation Form (scores ONLY):*

*- Engagingness:*

## Evaluate Hallucinations

*Human Evaluation of Text Summarization Systems:*

*Factual Consistency: Does the summary untruthful or misleading facts*

*that are not supported by the source text?*

*Source Text:*

*{{Document}}*

*Summary:*

*{{Summary}}*

*Does the summary contain factual inconsistency?*

*Answer:*