# OpenReview forum: "G-Eval: NLG Evaluation using Gpt-4 with Better Human Alignment"
_EMNLP/2023/Conference — EMNLP 2023 Main_

### Official Review · Reviewer_qnnB · 2023-08-04

**Typos Grammar Style And Presentation Improvements:** n/a
**Soundness:** 4

**Excitement:**

4: Strong: This paper deepens the understanding of some phenomenon or lowers the barriers to an existing research direction.

**Justification For Ethical Concerns:**

no evident ethical concerns

**Paper Topic And Main Contributions:**

The manuscript addresses the topic of evaluating NLG systems.
It continues a recent trend of using LLMs for NLG evaluation.
The specific proposal in this paper is to use Chain-of-Thought (CoT) prompting as an intermediate step in the evaluations.
The authors use data from two NLG tasks - summarization and dialog generation, and demonstrate that their proposed evaluation system
achieves better correlations with human evaluations than classic approaches like ROUGE or recent LLM-based approaches.


**Questions For The Authors:**

1.

Lines 38-40: "but they have been shown to have relatively low correlation with human judgments, especially for open-ended generation tasks".  What are the typical correlations for those? Maybe provide  references to papers.

2.

For the experiments conducted, human evaluations were used from previous research.
What scales were used by human evaluators?  1-5 ?
Same scales for all dimensions of evaluation?
This should be explicitly described in the current paper.

3.

When comparing human and  G-EVAL-without-probability-normalization (i.e. just the integer rating scales),
you should try  the  QWK metric (quadratic weighted kappa),
This measure of agreement is often used in Automated Essay Scoring literature for comparing human and automated scores.
It will also 'control' for chance agreement.





**Reasons To Accept:**

A good and creative idea and well-designed experimental work.

**Reasons To Reject:**

n/a

**Reproducibility:**

4: Could mostly reproduce the results, but there may be some variation because of sample variance or minor variations in their interpretation of the protocol or method.

**Reviewer Confidence:**

4: Quite sure. I tried to check the important points carefully. It's unlikely, though conceivable, that I missed something that should affect my ratings.

---

> ### Author Rebuttal · Authors · 2023-08-29
>
> Dear Reviewer,
>
> Thank you for your constructive feedback and positive comments on our paper. We appreciate your time and effort in reviewing our work. We would like to address your questions and concerns as follows:
>
> 1. Regarding your question about the typical correlations for open-ended generation tasks, the correlations vary depending on the specific task and evaluation metric used. For instance, UniEval reported a correlation of 0.128 between ROUGE-L and human evaluation for summarization tasks, which is usually considered as low correlation. We will add these references and more specific information in the revised manuscript to provide a clearer context.
>
> 2. Concerning the scales used by human evaluators in the experiments, SummEval used a scale of 1-5, TopicalChat used a scale of 1-3, Fact-based Benchmark used a scale of 0-1. We will explicitly describe this in the revised manuscript to avoid any confusion.
>
> 3. We appreciate your suggestion to use the QWK metric. We agree that this measure could provide a more robust comparison by controlling for chance agreement. We will incorporate this metric in our revised manuscript and discuss the results.

---

### Official Review · Reviewer_EnEs · 2023-08-05

**Soundness:** 4

**Excitement:**

3: Ambivalent: It has merits (e.g., it reports state-of-the-art results, the idea is nice), but there are key weaknesses (e.g., it describes incremental work), and it can significantly benefit from another round of revision. However, I won't object to accepting it if my co-reviewers champion it.

**Paper Topic And Main Contributions:**

This paper proposes an evaluation framework for models that generate text. The proposed evaluation framework, GEval, uses a LLM to first generate intermediate evaluation steps based on the task description. Then, the LLM's output is appended to a fixed prompt, and then has input context and the target text appended to it, with only a simple form that just asks for numerical scores after that. (I *think* this is right, please correct me if my read of this part of the paper got something wrong.) Different token output options for numerical scores by the LLM are then combined in a weighted average to produce a final score.

For the paper's experiments, they use GPT3.5 and GPT4 as the LLM for GEval, and they compare to a variety of baselines, including GPTScore. They evaluate on summarization and dialogue generation datasets, using correlation with human judgments as the metric of interest, finding that GEval-4 generally performs best.

The paper then performs an analysis of whether LLMs seem to have a bias towards preferring LLM-generated text, finding that this is indeed somewhat of an issue. There's then a discussion of how to interpret some of the Kendall-Tau based results reported earlier.

**Questions For The Authors:**

(A) Does the evaluating LLM ever actually have to perform the different evaluation steps that it generates? If not, this feels somewhat different from the original Chain of Thoughts setup. These are also pretty high-level steps compared to typical CoT... Am I missing something?

(B) Why are some of the listed baselines not shown in table 2?

**Reasons To Accept:**

A lot of the analysis in section 4 is strong and provides some really useful insights that future work can build off of, particularly the experiments that examine the cases in which LLMs seem to prefer LLM text to human-written texts. (If I had to nitpick, the section could perhaps do with an acknowledgment that aspects of how the human-written data was gathered may have also contributed to relatively low quality would be useful to add as a caveat, but the analysis is already useful as is.) The discussion of how the formulation of the Kendall-Tau correlation might affect the interpretation of results at first glance is also useful to note.

The proposed method (using GPT3.5 and GPT4) seems to outperform various existing baselines.

**Reasons To Reject:**

Given that the framework of GEval is LLM-agnostic, and GEval is the main contribution of the paper, it seems really important to compare GEval to at least one other text generation evaluator using the same LLM(s) in order to evaluate the benefits conferred by the framework itself, and I don't think that's currently happening? I understand that GPTScore requires information that's not currently given out by the OpenAI API for GPT4, but in that case, it seems important to evaluate GEval on the latest version of GPT that GPTScore can support as well, to allow for a fair comparison to GPTScore. I think that the reported GPTScore performance just uses GPT3, though, which GEval doesn't. As it is, I'm not sure how much of the improved correlation with human judgment is due to GEval, or just to the use of GPT3.5/GPT4.

**Reproducibility:**

4: Could mostly reproduce the results, but there may be some variation because of sample variance or minor variations in their interpretation of the protocol or method.

**Reviewer Confidence:**

3: Pretty sure, but there's a chance I missed something. Although I have a good feel for this area in general, I did not carefully check the paper's details, e.g., the math, experimental design, or novelty.

**Typos Grammar Style And Presentation Improvements:**

I think it's a bit of a stretch to claim that comparing results from GPT4 to those from GPT3.5 tells you "the effect of model size" (line 456). Since not every detail of the training process for OpenAI's models is published, there are enough unknown differences between the two models to make the comparison between these two models not necessarily representative of what happens as you vary model size in general.

The described setup doesn't quite seem like Chain of Thought to me? But this depends on the answer to question (A), I wasn't left completely sure about that.

---

> ### Author Rebuttal · Authors · 2023-08-28
>
> Dear Reviewer,
>
> Thank you for your insightful comments and suggestions. We appreciate your time and effort in reviewing our paper. We would like to address your concerns as follows:
>
> 1. Comparison with other text generation evaluators: We agree with your point that a comparison with other evaluators using the same LLMs would provide a more comprehensive evaluation of GEval. But we want to emphasize due to the API limit, GPT 3.5 is already the “the latest version of GPT that GPTScore can support”. And for a more detailed ablation of G-Eval we did a simple prompting baseline on SummEval (only asking GPT-4 to score a summary from 1-5 on each dimension, without detailed description and CoT)
>
> |  | Coh. ($\rho$)  | Coh. ($\tau$)  | Con. ($\rho$) | Con. ($\tau$)  | Flu. ($\rho$) |Flu.    ($\tau$)   | Rel. ($\rho$) | Rel .($\tau$)  |
> |:---:|:---:|:---:|:---:|:---:|:---:|:---:|:---:|:---:|
> | G-EVAL-4 | 0.582 | 0.457 | 0.507 | 0.425 | 0.455 | 0.378 | 0.547 | 0.433 |
> | GPT-4(simple prompt) | 0.513 | 0.424 | 0.421 | 0.344 | 0.447 | 0.373 | 0.479 | 0.388 |
>
> We will include these results in the revised manuscript to provide a more comprehensive comparison.
>
>
> 2. The role of the evaluating LLM: The evaluating LLM does not need to explicitly perform the different evaluation steps that it generates. The steps are used as a guide to structure the evaluation process and ensure that all relevant aspects of the task are considered. This can be considered a simple version of CoT with only one step. We will clarify this point in the revised manuscript.
>
> 3. Baselines not shown in Table 2: Table 2 is for Topical-Chat benchmark which evaluates dialogue generation tasks, and some baselines are designed only for evaluating summarization models. We have double-checked and all dialogue-related baselines are included in Table 2. We will clarify this in the next version.
>
> 4. Regarding the claim about the effect of model size: We agree that the comparison between GPT3.5 and GPT4 may not fully represent the effect of model size due to potential differences in the training process. We will revise this statement to more accurately reflect the limitations of our comparison.
>
> We hope that our responses address your concerns and we hope that you will reconsider the paper. We are dedicated to making improvements to the paper based on your valuable input.

---

### Official Review · Reviewer_LBER · 2023-08-11

**Typos Grammar Style And Presentation Improvements:** 1. There is an error in the third sub…
**Soundness:** 2

**Excitement:**

3: Ambivalent: It has merits (e.g., it reports state-of-the-art results, the idea is nice), but there are key weaknesses (e.g., it describes incremental work), and it can significantly benefit from another round of revision. However, I won't object to accepting it if my co-reviewers champion it.

**Justification For Ethical Concerns:**

None.

**Missing References:**

None.

**Paper Topic And Main Contributions:**

G-EVAL delves into the challenges of evaluating the quality of text generated by NLG systems. Traditional metrics, such as BLEU and ROUGE, often have low correlation with human judgment, especially in tasks that require creativity and diversity in generated text. This paper focuses on enhancing the evaluation metrics of NLG systems. The proposed G-EVAL framework is an innovation with excellent performance and promises for more consistent and accurate evaluation of generated text.

The main contributions of this paper are:
1. G-EVAL framework: The author introduces the G-EVAL framework, a method that adopts LLM as an evaluation indicator, especially with the "thought chain" and form filling paradigm. This more effectively evaluates the quality of NLG output.
2. Multiple task experiments: This study conducted extensive experiments on tasks such as text summarization and dialogue generation, demonstrating the versatility of the method.
3. Excellent performance: The performance of the G-EVAL with GPT-4 as the backbone surpasses the previous method on three tasks.
4. LLM-based evaluator analysis: Section 4 of this paper will provide an interesting finding on the potential bias of LLM-generated text.

**Questions For The Authors:**

First, I commend the authors for their innovative approach utilizing GPT-4. The paper is well structured and the contributions are interesting. However, to fully understand the scope, impact and future direction of this research, I have a few questions:

1. How do you avoid the instability or unreliability of GPT-4 and GPT-3.5 as metrics? As far as I know, there have been such studies on other LLM fine-tuning methods.

2. I would like to see how much your method improves on a GPT-4 baseline that doesn't use your method at all, not just compared to other metrics (like BertScore, Rouge......), because I don't know if the performance improvement is from GPT-4 or your method. Can you show me this result? Just need a simplest prompting when using GPT-4.

**Reasons To Accept:**

This article provides new perspectives and methods for the long-standing problem of how to evaluate the output of natural language generation systems, which is of great significance for the widespread use of current NLG applications.

1. Innovative approach: The G-EVAL framework utilizes large language models with "CoT and form-filling paradigms", which demonstrate better evaluation than traditional metrics such as BLEU and ROUGE. G-EVAL has potential applications across multiple NLG tasks, such as text summarization and dialogue generation.

2. Extensive experimental verification: The paper conducts extensive experiments, which compares it with existing methods and shows its superiority.

3. Discussion of potential problems: In addition to proposing new methods, the author also analyzes the problems that may be encountered when using LLM as an evaluator, which provides an important reference for related research.

Benefits to the NLP community:

Provide a new evaluation tool: G-EVAL can provide the NLP community with a new NLG evaluation tool.

**Reasons To Reject:**

Although this article uses GPT-4 as a new metric for NLG evaluation, two questions make me question:

1. The unreliability of GPT-4 is not considered:
As a metric for evaluation, the stability of the results must be guaranteed, that is, the same dataset, when using this metric at any time (such as the past three months, now, next year...) is guaranteed to obtain the same results. Obviously, I don't see this in your method, because some recent studies have shown that GPT-4 is gradually getting stupid (https://arxiv.org/pdf/2307.09009.pdf ,  https://github.com/THU-KEG/ChatLog ), which means that when using GPT-4 in the text as the skeleton and using the same parameters, I can't reproduce your results after a while, then it loses the meaning of the metric. Of course one solution is to use other LLMs for fine-tuning as the skeleton, but I don't see this in this paper.

2. Lack of baseline comparison: The G-EVAL method in this paper is entirely based on GPT-3.5 and GPT-4, but does not demonstrate their performance when using the simplest prompt for these two LLMs, and then compare it with the G-EVAL method.

**Reproducibility:**

1: Could not reproduce the results here no matter how hard they tried.

**Reviewer Confidence:**

5: Positive that my evaluation is correct. I read the paper very carefully and I am very familiar with related work.

---

> ### Author Rebuttal · Authors · 2023-08-28
>
> Dear Reviewer,
>
> Thank you for your insightful comments and constructive feedback. We would like to address your concerns as follows:
>
> 1. Unreliability of GPT-4: We acknowledge your concern about the potential instability of GPT-4. However, we believe that the "gradual degradation" of GPT-4 is not a significant issue for our method. The G-EVAL framework is designed to be adaptable to different LLMs, and we chose GPT-4 for our experiments due to its state-of-the-art performance. If GPT-4's performance degrades over time, we can easily switch to a different LLM. Meanwhile, we re-ran experiment with the latest GPT-4-0613 and get the results on SummEval:
>
> |  | Coh. ($\rho$)  | Coh. ($\tau$)  | Con. ($\rho$) | Con. ($\tau$)  | Flu. ($\rho$) |Flu.    ($\tau$)   | Rel. ($\rho$) | Rel .($\tau$)  |
> |:---:|:---:|:---:|:---:|:---:|:---:|:---:|:---:|:---:|
> | G-EVAL-4 | 0.582 | 0.457 | 0.507 | 0.425 | 0.455 | 0.378 | 0.547 | 0.433 |
> | G-EVAL-4(0613) | 0.593 | 0.462 | 0.508 | 0.425 | 0.465 | 0.382 | 0.545 | 0.430 |
>
> We can see that GPT-4 is not getting worse as an evaluator.  We will clarify this point in the revised manuscript. We will also report meta-evaluation results of different versions of GPT-4 to discuss this point.
>
> Meanwhile, we want to emphasize that "as a metric for evaluation, the stability of the results must be guaranteed" is only applied to conventional automatic metrics, but not to human evaluation (different human annotators will have different judges of the same output). And with more and more complex NLG tasks proposed, we believe frameworks like G-EVAL which is designed to mimic human evaluation, should not be constrained by this. Instead, we could use a pre-meta-evaluation test to make sure the evaluator's performance is reasonable. We appreciate your insightful comments on this point and will add more discussion in the limitation section.
>
>
> 2. Lack of baseline comparison: We agree that a comparison with a simple GPT-4 baseline would be beneficial. We did a simple prompting baseline on SummEval (only asking GPT-4 to score a summary from 1-5 on each dimension, without detailed description and CoT)
>
> |  | Coh. ($\rho$)  | Coh. ($\tau$)  | Con. ($\rho$) | Con. ($\tau$)  | Flu. ($\rho$) |Flu.    ($\tau$)   | Rel. ($\rho$) | Rel .($\tau$)  |
> |:---:|:---:|:---:|:---:|:---:|:---:|:---:|:---:|:---:|
> | G-EVAL-4 | 0.582 | 0.457 | 0.507 | 0.425 | 0.455 | 0.378 | 0.547 | 0.433 |
> | GPT-4(simple prompt) | 0.513 | 0.424 | 0.421 | 0.344 | 0.447 | 0.373 | 0.479 | 0.388 |
>
> We will include these results in the revised manuscript to provide a more comprehensive comparison.
>
> 3. Regarding the error in Figure 1, we appreciate your attention to detail and will correct this in the revised version.
>
> We believe that our responses address your concerns and we hope that you will reconsider the paper. We are dedicated to making improvements to the paper based on your valuable input.

---

### Meta-Review · Area_Chair_imZw · 2023-09-15

**Recommendation:** 4

**Metareview:**

The reviewers are generally in agreement about the quality of the idea and execution of the experiments in this paper. The one point of disagreement is one reviewer's concern about the unreliability of GPT-based metrics; however (as another reviewer and the authors point out) reliability is a challenge across many of today's evaluation practices (including human evaluations). The authors agreed to discuss this topic in more depth, particularly in the limitations section, which I think is sufficient, given the scope of the paper.

---

### Decision · Program_Chairs · 2023-10-07

**Decision:**

Accept-Main

**Comment:**

The reviewers are generally in agreement about the quality of the idea and execution of the experiments in this paper. The one point of disagreement is one reviewer's concern about the unreliability of GPT-based metrics; however (as another reviewer and the authors point out) reliability is a challenge across many of today's evaluation practices (including human evaluations). The authors agreed to discuss this topic in more depth, particularly in the limitations section, which I think is sufficient, given the scope of the paper.